# Comparison of Key Characteristics of Remarkable SSW Events in the Southern and Northern Hemisphere

**Michal Kozubek \*** **, Jan Lastovicka** and **Peter Krizan**

Institute of Atmospheric Physics of the Czech Academy of Sciences, 14100 Prague, Czech Republic;
jla@ufa.cas.cz (J.L.); krizan@ufa.cas.cz (P.K.)
**\*** Correspondence: kom@ufa.cas.cz

**Abstract:** An exceptionally strong sudden stratospheric warming (SSW) in the Southern Hemisphere (SH) during September 2019 was observed. Because SSW in the SH is very rare, comparison with the only recorded major SH SSW is done. According to World Meteorological Organization (WMO) definition, the SSW in 2019 has to be classified as minor. The cause of SSW in 2002 was very strong activity of stationary planetary wave with zonal wave-number (ZW) 2, which reached its maximum when the polar vortex split into two circulations with polar temperature enhancement by 30 K/week and it penetrated deeply to the lower stratosphere and upper troposphere. On the other hand, the minor SSW in 2019 involved an exceptionally strong wave-1 planetary wave and a large polar temperature enhancement by 50.8 K/week, but it affected mainly the middle and upper stratosphere. The strongest SSW in the Northern Hemisphere was observed in 2009. This study provides comparison of two strongest SSW in the SH and the strongest SSW in the NH to show difference between two hemispheres and possible impact to the lower or higher layers.

**Keywords:** sudden stratospheric warmings; reanalysis; stratospheric dynamics

## 1. Introduction

Sudden stratospheric warmings (SSWs) are one of the most impressive dynamical events in the stratosphere. These events include a large and rapid temperature increase (>30–40 K on time-scale of days) in the mid- to upper stratosphere (30–50 km) and, in the cases called major, a reversal of the climatological westerly zonal-mean zonal winds associated with the stratospheric polar night jet (e.g., [1–3]). They are usually driven by the breaking of planetary waves from the troposphere in the stratosphere.

There are many definitions for SSW type (major, minor, final, and Canadian) in literature and their summary can be found in [4]. The basic definition of the major SSW approved by WMO (World Meteorological Organization) is as follows: A stratospheric warming can be said to be major if at 10 hPa or below the latitudinal mean temperature increases poleward from 60 degrees latitude and an associated circulation reversal is observed (i.e., zonal westerly winds poleward of 60° latitude are replaced by mean easterlies in the same area). There is a broad discussion in atmospheric community which definition is the best one, but we used the WMO definition for our study. SSWs may be defined and or grouped in different ways according to purpose of study (e.g., [5]). Next important point of major SSW event definition is a duration of easterly zonal-mean zonal winds because the zonal mean zonal wind reversal could be short (about 1 day) and very weak (with values only slightly below zero). Reference [6] show that difference between SSW major and SSW minor events as is defined according to WMO could eliminate events which have strong impact on atmospheric dynamics.

SSWs are very important because temperature and wind anomalies associated with them can descend downward into the troposphere on time scales of weeks to months and they can have

significant impacts on wintertime surface climate on both hemispheres [7]. The possible impact of SSWs on troposphere could be closely connected with the negative phase of the North Atlantic Oscillation (NAO) with an equatorward shift of the North Atlantic storm track; extreme cold air outbreaks in parts of North America, northern Eurasia, and Siberia (e.g., [8]). SSWs are also connected with other phenomena like quasi-biennial oscillation (QBO), El Niño–Southern Oscillation (ENSO) or solar cycle. This connection was studied for example in papers by [9,10]. Dynamical effects of planetary wave (PW) breaking during SSWs are observed not only in the stratosphere but we can detect them in the mesosphere and lower thermosphere as well [11,12]. Effects of SSWs were detected even in the ionosphere (e.g., [13]). Several papers also studied predictability of SSW or comparison of SSW with model output (e.g., [14]).

Major midwinter SSWs rarely occur in the Southern Hemisphere, mainly because of weaker planetary-wave amplitudes [15]. The only observed exception occurred in September 2002, when a major SSW occurred. Many studies describe the dynamics and ozone hole problem in the middle atmosphere, before and during this event (e.g., [16–18]). Another strong SSW on the Southern Hemisphere occurred in 2019 (details of observations can be find in [19,20]). This SSW is one of the strongest on the SH and that is why it is important to compare its main characteristic with the major SSW in 2002 to identify similarity between these two events for future possible studies not only in the stratosphere but also in the ionosphere or troposphere. Paper [21] describes effect of SSW in 2019 on the low latitude ionosphere based on Swarm data.

## 2. Data and Methods

We use ECMWF (European Center for Medium-Range Weather Forecasts) reanalysis ERA5, which detailed description can be found in ERA5 data documentation or in [22], available online: https://software.ecmwf.int/wiki/display/CKB/ERA5+data+ documentation (accessed on 18 July 2018). Data are downloaded from ERA 5 [23]. The ERA5 is available for the period from 1980 till present on hourly basis but here we use 00, 06, 12, and 18 UTC (Coordinated Universal Time) values for each parameter (temperature, zonal wind, or geopotential). ERA-5 has the resolution $0.75° \times 0.75°$.

SSW used to be characterized mainly by temperature, wind, and geopotential changes in the middle stratosphere, particularly at 10 hPa. That is why we analyzed connections between polar temperature, zonal wind (and its reversal from westerly to easterly at 60° N or S), and geopotential from 1 to approximately 100 hPa. We studied zonal wind especially on the 60° latitude, which correspond with definition of SSW according to WMO. We analyzed period approximately one month before and after each SSW to show antecedent and recovery behavior of the stratosphere and behavior during SSWs. The major and minor SSWs on the SH (2002 and 2019) and on the NH (2009) are analyzed in the paper.

We analyzed polar temperature and zonal wind 60° N/S at 3 pressure levels (7, 10, and 30 hPa), where we can expect the biggest effect of SSW in the stratosphere. Then, we analyzed the vertical temperature profile (from 1 to 100 hPa) for latitudes 90°–50° N/S of each major SSW during the whole SSW. We also computed zonal averages of temperature for 85° N/S and zonal wind for 60° N/S at 1 and 10 hPa for 3 winter months. Finally, we analyzed behavior of geopotential at 10 hPa during each major SSW.

The focus of this study is mainly on comparison of the major SSW in 2002, SSW in 2019 in the SH, and strong major SSW in 2009 in the NH. We also show analysis of minor or no SSW years for both hemispheres. Special attention is paid to the SSW of 2019, which is classified as a minor one according to the WMO definition. On the other hand, the polar temperature increase between 5 September and 11 September reached about 50 K/week, which is much stronger increase than that for SSW in 2002, so we speculate if the WMO definition is sufficient. That is why we will show other characteristics, which show us more differences between these two events, and comparison with other SSWs.

## 3. Results

Temperature and zonal wind are two of the main characteristics for dynamics during the polar night on both hemispheres. Figures 1–3 show polar temperature (90° N or S) and zonal wind at 60° N or S for period 1 August–31 October 2002, and 2019 in the SH and 1 January–31 March 2009 in the NH. These periods represent the behavior of the temperature during all studied SSWs. The period between 1 January and 31 March represents the majority of observed SSWs in the NH. We present results for both pressure levels (1 and 10 hPa), where we can expect a strong effect of stratospheric warming. The top panels show polar temperature during the whole period of each SSW, which reveal the main characteristic of SSW (beginning, maximum, and end of SSW). The bottom panels show us zonal wind at 60° N or S, which is important for classification of SSWs.

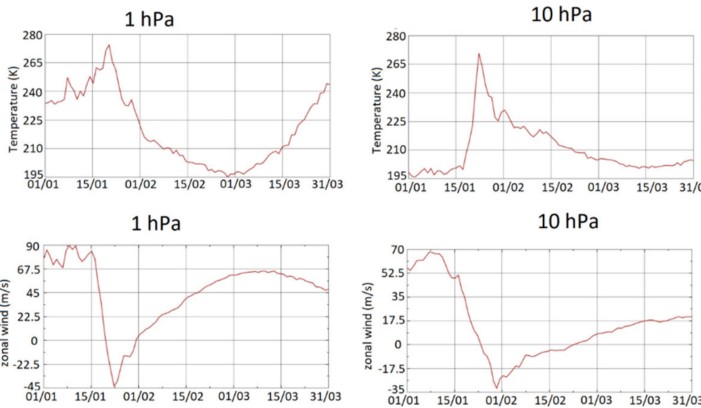

**Figure 1.** Polar temperature at 90° N for 1 and 10 hPa (**upper panels**) and zonal mean of zonal wind at 60° N (**bottom panels**) for 1 and 10 hPa during 1 January 2009–31 March 2009.

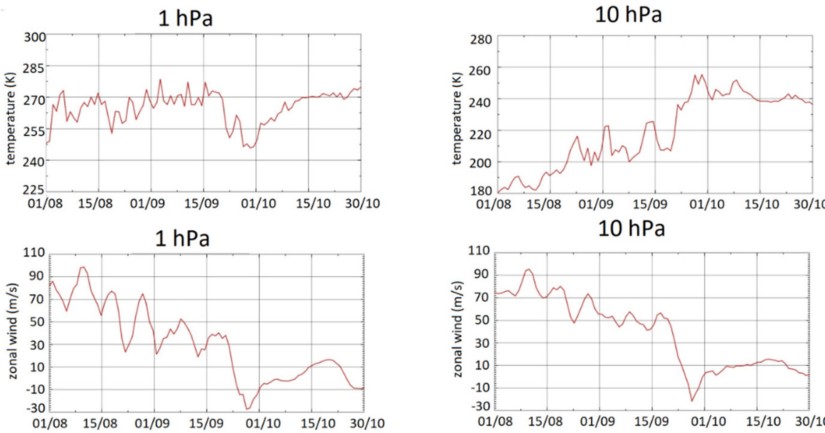

**Figure 2.** The same as Figure 1 but for the Southern Hemisphere (SH) and period 1 August 2002–30 October 2002.

Figure 1 shows major SSW in the NH in 2009. It started in mid-January (20 January) with a strong increase of polar temperature (at about 60 K in a week), whereas strong zonal wind reversal at 10 hPa started on 24 January. The maximum temperature occurred on 23–24 January when temperature at 10 hPa reached 270 K, whereas the well-pronounced maximum of easterly wind occurred on 28 January. The end of SSW in temperature is difficult to determine, it was late February to early March, while the zonal wind returned to westerly on 22 February. At 1 hPa we can see that the maximum of SSW occurred several days earlier, which confirms theory that SSW starts in the upper stratosphere/lower mesosphere and descends during several days to the middle stratosphere.

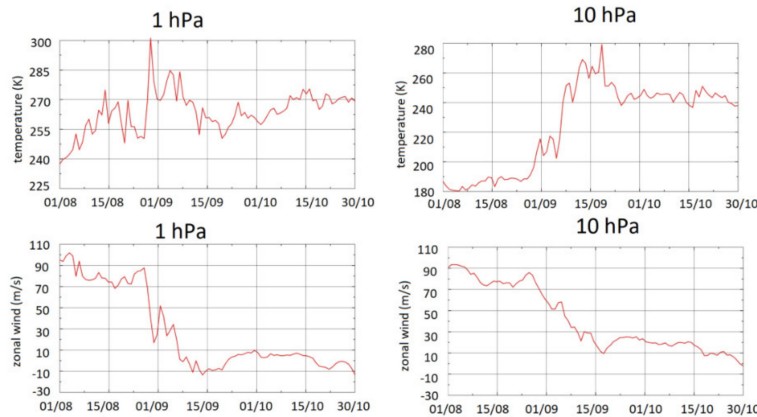

**Figure 3.** The same as Figure 2 but for the period 1 August 2019–30 October 2019.

Figure 2 shows major SSW in the SH in 2002. This is the only observed major SSW in the SH. It started in late September (21 September) with an increase of polar temperature (at about 40 K in a week), whereas zonal wind reversal at 10 hPa started on 24 September. The maximum temperature occurred on 29–30 September when temperature at 10 hPa reached 255 K, whereas well-pronounced maximum of easterly wind occurred on 27 September. The end of SSW in temperature is difficult to determine because higher temperature remained for several weeks, while the zonal wind returned to westerly on 1 October but remained very weak in comparison with the pre-SSW period. At 1 hPa the influence of SSW on temperature was not so pronounced as at 10 hPa and only a weak disturbance during maximum can be observed. On the other hand, the zonal wind reversal was as strong as at 10 hPa.

Figure 3 shows SSW in the SH in 2019. It started at the beginning of September 2019 (6 September) with an increase of polar temperature (by about 65 K in a week). The wind reversal did not reach 10 hPa (it occurred only in the upper stratosphere), so we cannot classify it as a major SSW. However, wind reversal was observed at 1 hPa. The maximum of the SSW occurred on 18–20 September, when temperature at 10 hPa reached 280 K. This SSW was the strongest SSW in the SH and the temperature increase was stronger than during the major SSW in 2002. A substantial difference can be observed at 1 hPa, where an increase of temperature started almost 14 days earlier than at 10 hPa. It means that evolution is different from major SSW in 2002 or SSW in 2009.

We also analyzed three years with no major SSW for comparison, noting that SH temperature and zonal wind at 10 hPa were very stable during the whole winter time (see winter 2015 in Figure S1). Several minor warmings with weak influence on zonal wind were observed in the NH, e.g., winter 2014 in Figure S2). At 10 hPa we can see three small warmings (up to 20 K in several days). This behavior is usual for NH, because polar vortex is disturbed by wave activity from troposphere. The last year we analyzed was winter 2020, when an unusual strong vortex and cold temperature occurred and remained until late March (Figure S3). Strong ozone depletion connected with this feature was observed.

Until now, we studied only zonal average of polar temperature or zonal wind to identify if major SSW occurred or not. Next, we analyzed latitude vs. time development for all studied years. The temperature and zonal wind evolution at three pressure levels (7, 10, and 30 hPa) are shown in Figures 4–6. They represent middle stratosphere, which influence not only the upper stratosphere but the lower stratosphere and tropopause as well. Figure 4 shows winter 2009 in the NH. We can identify strong warming at 7 and 10 hPa, and even at 30 hPa it is still detectable. This major SSW was the strongest between 90° and 65° N, but it can be detected down to 50° N at 10 hPa. Zonal wind behavior was much more pronounced than temperature changes. It is observed especially south from 75° N. The temperature changes are detectable several weeks after the maximum of SSW. The strongest effect was around 60° N but we can detect changes connected with this SSW at 50° N maybe lower at 30 hPa. This means that SSW affects dynamics in the lower stratosphere and mid-low latitudes as well.

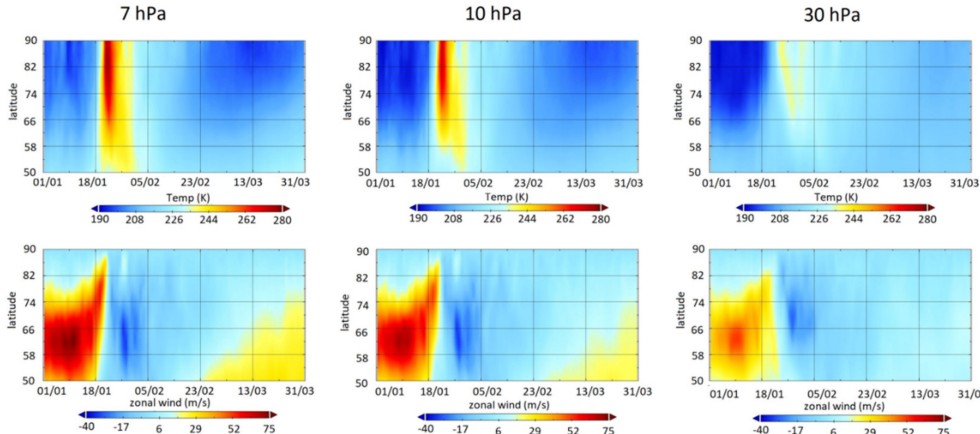

**Figure 4.** Zonal mean of temperature (**upper panels**) and zonal wind (**bottom panels**) for 7, 10, and 30 hPa during period 1 January 2009–31 March 2009.

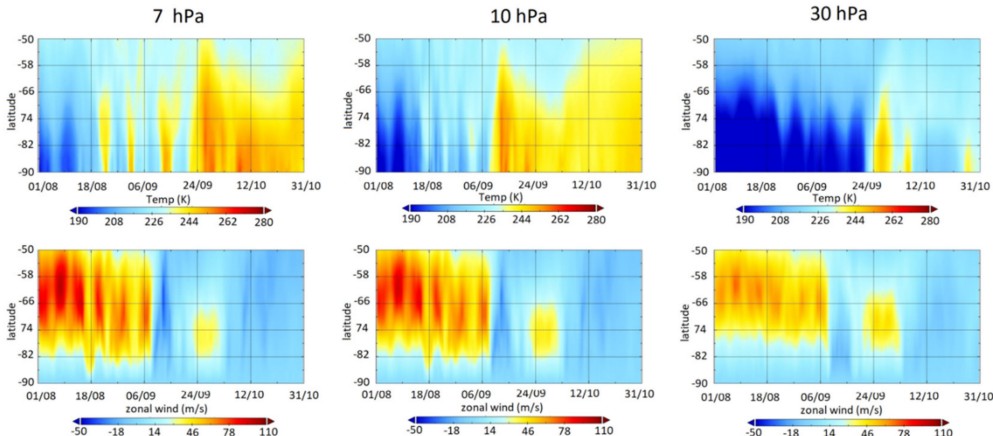

**Figure 5.** The same as Figure 4 but for the SH and period 1 August 2002–30 October 2002.

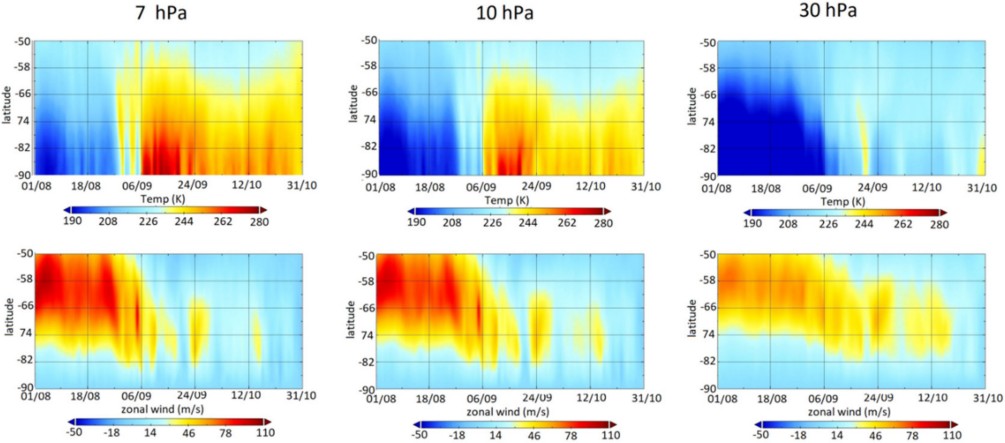

**Figure 6.** The same as Figure 5 but for the period 1 August 2019–30 October 2019.

Figure 5 shows winter in 2002 in the SH. As pointed out before, this was the only major SSW identified in the SH. We see different behavior in comparison with NH SSW. Before this SSW, there were three weaker warmings, and higher temperature remained for several weeks at all pressure levels, which could be explained by the final SSW, which normally occurs in mid-October. The effect of SSW is visible down to 55° S at 7 and 10 hPa and down to 60° S at 30 hPa. Zonal wind reversal impact

is visible down to 50° S at all pressure levels with a short disturbance in late September. It is very interesting that warmings began after zonal wind reversal and not before at 30 hPa.

Figure 6 shows SSW in 2019 in the SH. This SSW, as was mentioned above, was minor, but changes in temperature and zonal wind at higher stratospheric layers were bigger than during SSW in 2002. We observe similar behavior to SSW in 2002 when two smaller warmings occurred before main SSW, and after this the effect of final warming can be seen. SSW is detectable down to 60° S at 7 and 10 hPa and 65° S at 30 hPa. We will show later that this SSW does not propagate so deep into stratosphere as SSW in 2002 or 2009. Zonal wind development is different to SSW in 2002 because we do not see wind reversal at 10 and 30 hPa. Only at 7 hPa is there detectable weak reversal. That is why we cannot classify this SSW as major. We analyzed another three years without major SSW in the same way (result shown in Supplement). Zonal wind behavior shows strong and undisturbed polar vortex during the whole winter 2015 (see Figure S4). During winter 2014 in the NH, we can identify several minor warmings, which were mentioned above, but zonal wind remained westward during whole winter (see Figure S5). Finally, during winter 2020 in the NH, we can see only weak warming, but temperature remained very low during the whole winter and especially zonal wind behavior shows very strong vortex even in April (see Figure S6).

Now we will focus in detail on the three warmings in 2002 and 2019 in the SH and 2009 in the NH. We will analyze development of geopotential height at 10 hPa during SSW and vertical profile of temperature as well. These characteristics are able to show us difference between SSW in the SH and NH and how they can influence lower part of the stratosphere or upper troposphere. Figure 7 shows temperature vertical profile for several days before, during, and after major SSW in 2009. We start on 20 January 2009 when strong temperature increase is firstly observed. On 24 January, when maximum of temperature was reached, an increase of polar region temperature penetrated down to 30 hPa and SSW was visible down to 60° N. Later the stratosphere was colder but higher temperature could be detected even at 180 hPa up to 75° N. This means that this SSW can influence lower stratosphere and upper troposphere at high latitudes even in mid-February, when zonal wind reversed back to normal. Figure 8 shows geopotential at 10 hPa during SSW. We can easily identify that the normal situation of 10 January 2009 changed on 20 January when SSW began. We also identify split of polar vortex on 24 January when the maximum temperature was reached. This situation remained until late February, so we can say that in terms of geopotential, this SSW ended in late February.

Figure 9 shows temperature vertical profile of major SSW in 2002 in the SH. It starts on 17 September 2002, when normal situation in stratosphere occurs. On 21 September we detected strong temperature increase but vertical profile remains of normal shape. However, during maximum of SSW on 29 September, the SSW penetrated down to 80 hPa. This was also the lowest level which is reached during this SSW and after that it began to normalize. The end of SSW was in mid-October. Here we can see the difference between major SSW in 2009 and 2002. During maximum of SSW in 2009, substantial heating penetrated down to 200 hPa, while during the maximum of SSW in 2002, the warm part reached from 80 and possibly up to 100hPa (23 September–1 October), and the lowest part of stratosphere and the upper troposphere were not significantly affected. This finding is supported also by Figure 10, which shows geopotential at 10 hPa. On 17 September, the polar vortex was going to be disturbed and during the maximum was split into two cells. However, it was quickly restored and undisturbed situation is visible in mid-October. If we compare this behavior with SSW in 2009, the normal situation can be seen not earlier than at least one month after maximum of SSW. This can be explained by different dynamics on both hemispheres, where the SH planetary waves, which are mainly responsible for SSW, are not generated so easily in the lower part of the atmosphere (troposphere), while in the NH, these waves have several sources in the troposphere.

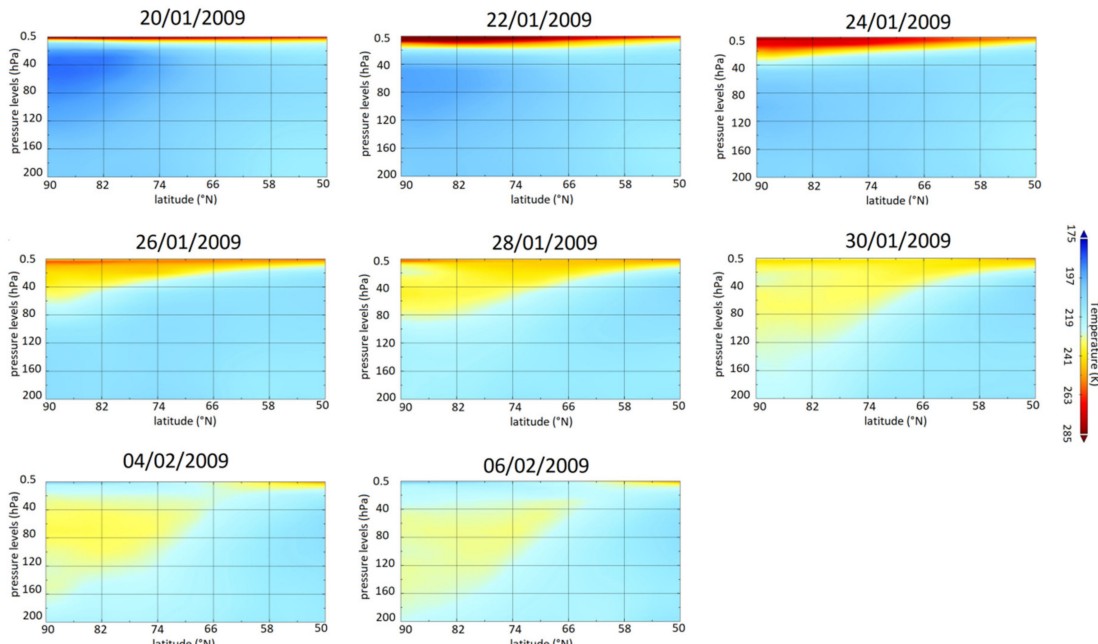

**Figure 7.** Vertical profile of polar temperature during sudden stratospheric warming (SSW) in 2009 in the Northern Hemisphere (NH).

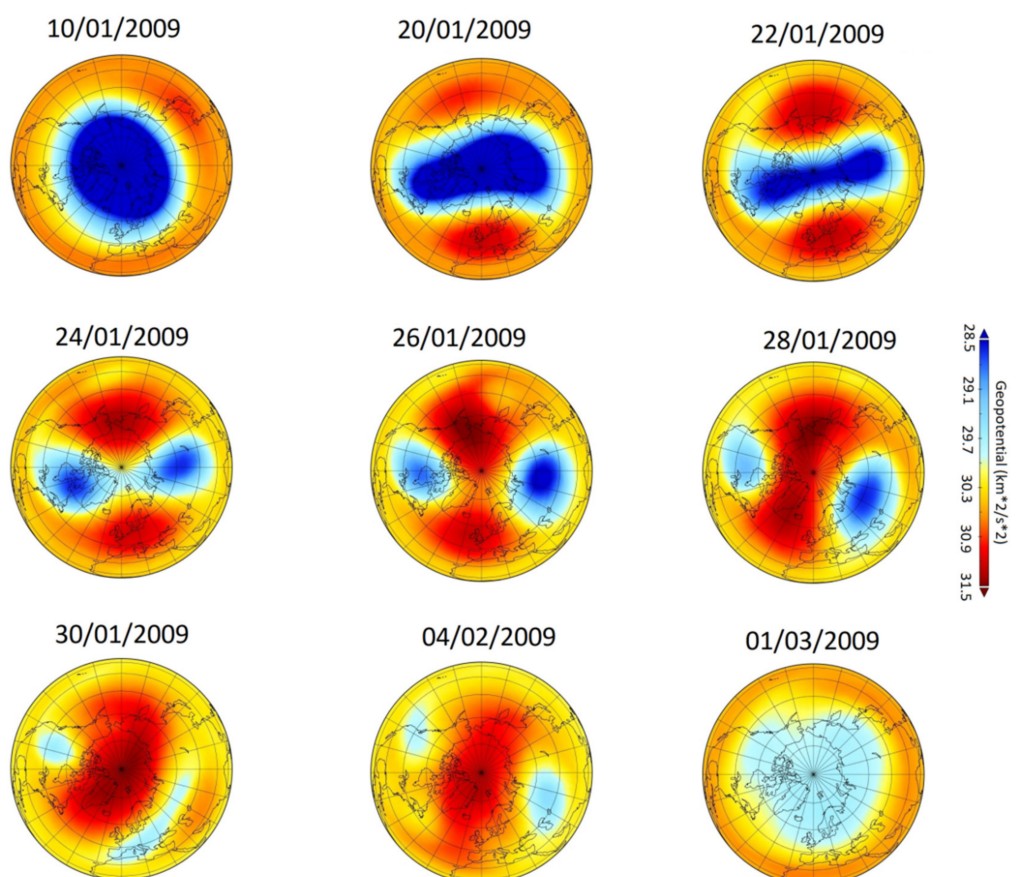

**Figure 8.** Geopotential heights at 10 hPa during SSW in 2009 in the NH.

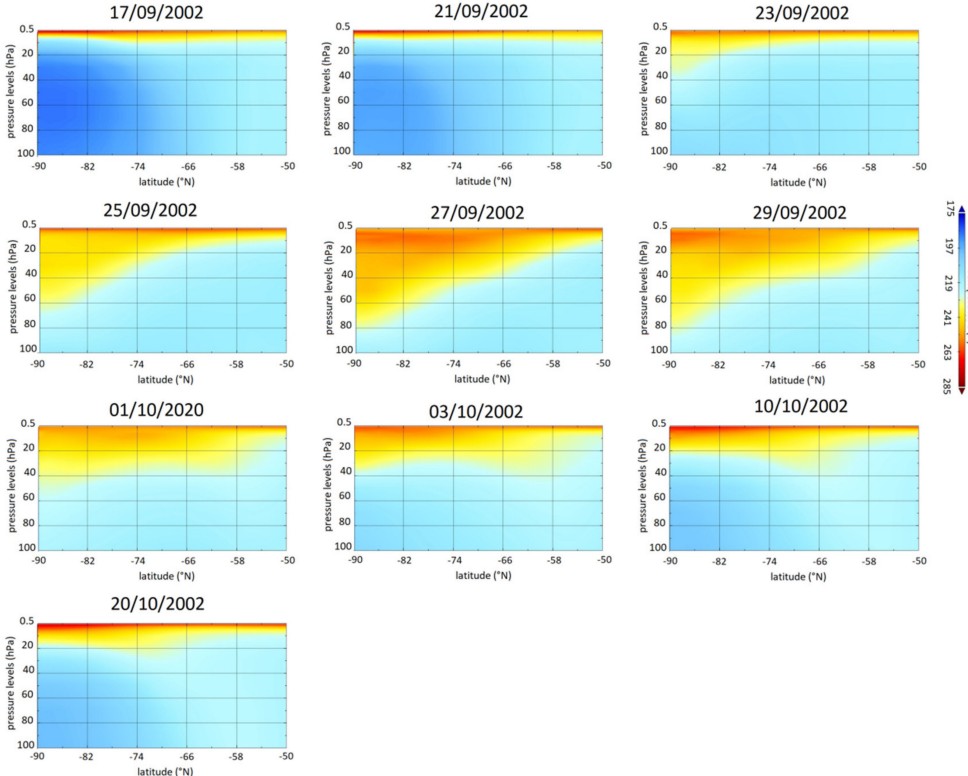

**Figure 9.** Vertical profile of polar temperature during SSW in 2002 in the SH.

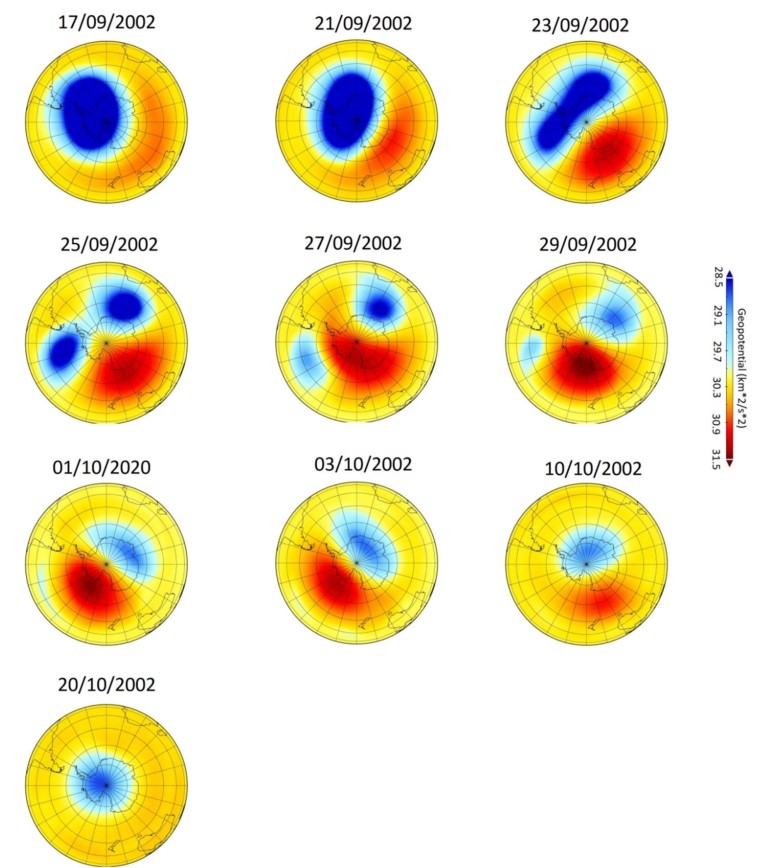

**Figure 10.** Geopotential heights at 10 hPa during SSW in 2002 in the SH.

The last SSW we focus on is in September 2019 in the SH. This SSW is unusually strong as was mentioned before but it cannot be classified as a major SSW. Figure 11 shows vertical temperature profile. The SSW can propagate down to 40 hPa and its maximum was on 18 September. At the beginning and after maximum, it affects only the upper and middle stratosphere above and around 20 hPa. That is very different from major SSW in 2002, even though the temperature increase is much bigger. The feature can be observed on Figure 12, which displays geopotential at 10 hPa. We can see that polar vortex is displaced in maximum of SSW but still it remains near the pole and the restoration of situation is very quick (on 24 September we almost cannot see any difference from normal situation). So, we can only speculate what processes are responsible for this SSW.

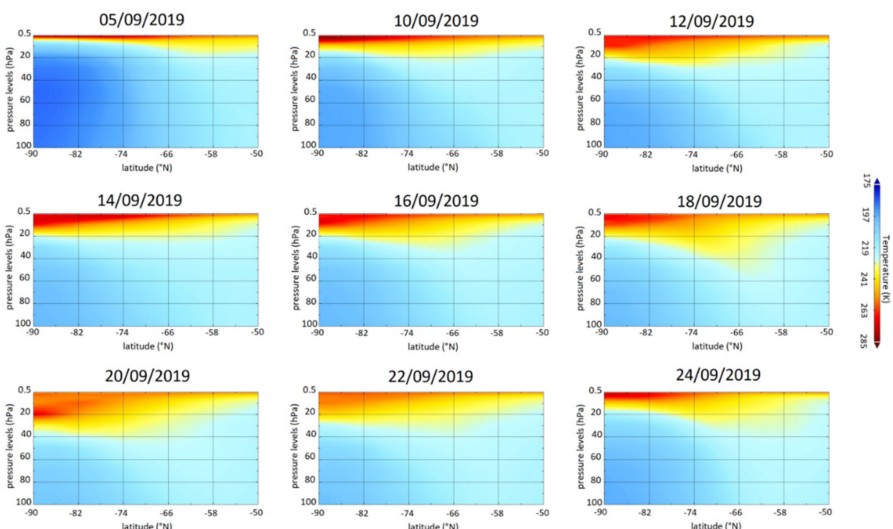

**Figure 11.** Vertical profile of polar temperature during SSW in 2019 in the SH.

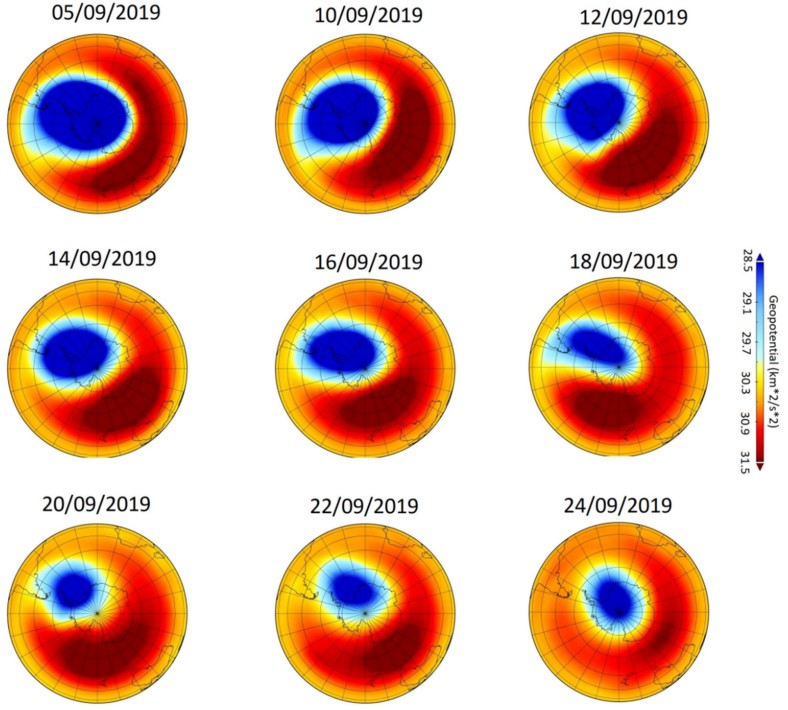

**Figure 12.** Geopotential heights at 10 hPa during SSW in 2019 in the SH.

## 4. Discussion

According to [18], the plausible cause of SSWs is very strong activity of the stationary planetary wave with zonal wave-number (ZW) 1 or 2, which reach maximum when the polar vortex split into two circulations and this is confirmed by the other studies, e.g., [16]. We have shown that even if the split of polar vortex is observed for SSW on both hemispheres (2009 and 2002), the situation during, and mainly after, the maximum of SSW is different. We noticed that the SSW in 2009 was a very strong SSW observed in the NH. This has to be taken into account during the comparison with SSW in the SH. On the other hand, the maximum increase in temperature during SSW 2009 and SSW 2019 is almost comparable. The return to normal situation was much quicker in the SH and only the middle stratosphere (down to ~70 hPa) was affected, whereas in the NH, we can see effect of SSW several weeks after maximum and it influenced not only stratosphere but upper troposphere as well. Karpechko et al. [24] showed that the tropospheric impact can be detected for up to two months, which is in agreement with our study. This difference could be caused by different behavior of zonal wind, which is much stronger in the SH during winter time, and that is why planetary waves, which are responsible for SSWs, are trapped and dissipate quicker than in the NH. This agrees with other results [25–27] that showed that the planetary waves have smaller amplitudes in the SH than in the NH because of less land-sea contrasts and smaller topographical differences. According to [21], an exceptionally strong stationary planetary wave with ZW 1 was observed during September 2019. That situation brought a very strong temperature increase but without zonal wind reversal at 10 hPa. It should be mentioned that the stratopause breakdown and subsequent reformation at very high altitude, accompanied by enhanced descent into a rapidly strengthening upper stratospheric vortex, occurs with many major SSWs; for example, in 2009 and 2006 [28]. The SSWs warming is typically accompanied by mesospheric cooling at higher levels [29–31]. Unfortunately, there are not so many studies which analyze higher atmospheric layers like mesosphere or ionosphere. That is why we cannot compare results from SSW 2002 and 2019. One of the studies, which analyzed mesospheric conditions during SSW in 2002 is [16]. They identified 14-day oscillation at about 80 km. Chandran et al. [11] showed interaction between stratosphere and mesosphere during SSWs. They also found that during normal conditions the polar stratopause is a gravity wave driven phenomenon but during SSW events it is mainly a planetary wave driven phenomenon. That is why it is very important to decide when SSW begins and ends and set appropriate definition of SSW, not only for stratosphere, but for mesosphere/ionosphere as well. Yamazaki et al. [21] described quasi-six-day wave activity for SSW in 2019, which was generated in the polar stratosphere at 40 km, so it could partly explain why temperature at 1 hPa was much more variable before and during SSW than during 2002, where main disturbances came from lower levels [18].

Krüger et al. [32] demonstrated the relevance of eastward-traveling waves in preconditioning the SH 2002 major warming event. They pointed out that stronger, and more frequent eastward-traveling height wavenumber-2 events can occur in a changing climate. A stronger vertical temperature gradient and related strong PNJ could lead to an enhanced forcing of such eastward-traveling height wavenumber-2 events in both hemispheres. Dowdy et al. [16] showed that quasi-10-day wave of wavenumber s = 1 travelling in an eastward direction was identified as responsible for triggering the occurrence of the 2002 major SSW. According to [33] in the stratosphere there was the coherent persistence of a traveling W2 throughout the whole winter. By September, two conditions are fulfilled that made the major warming possible. First, there were large traveling wave-2 amplitudes persisting on account of the wave geometry of the mean flow, and second, the vortex was weaker than normal.

Yamazaki et al. [21] shows that an exceptionally strong stationary planetary wave with ZW 1 led to SSW in the SH in September 2019. Ionospheric data show prominent six-day variations in the dayside low-latitude region at this time, which can be attributed to forcing from the middle atmosphere by the Rossby normal mode "quasi-6-day wave" (Q6DW). The Q6DW is apparently generated in the polar stratosphere at 30–40 km, where the atmosphere is unstable due to strong vertical wind shear

connected with planetary wave breaking. Comparison of wave activity during SSW 2002 and SSW 2019 shows that each event has different conditions before and during these SSWs.

On the other hand, if we focus on the ozone hole problem during these two SSWs, we can find some similarities. The existence of the very strong planetary waves in Antarctica in September 2002 led to the downward extension of the Antarctic ozone hole split up to the upper troposphere, which led to the no-ozone hole year in 2002 [34]. The anomalous geometry of the polar vortex (its tilt and decrease in size with altitude) accounted for all of the difference in the ozone hole area between 2018 and 2019 during the first half of September and more than half of the difference afterward. The area stayed between $5 \times 10^6$ and $10 \times 10^6$ km$^2$ in September and October 2019, compared to $20 \times 10^6$ and $25 \times 10^6$ km$^2$ in 2018 see [35]. Dynamical parameters suggest locally reversed and weakened zonal winds and a shift in the location of the polar jet vortex. This led to air masses mixing, to a reduced polar stratospheric clouds formation detected at a ground station, and as such to lower ozone depletion. 2019 total ozone columns for the months of September, October, and November were on average higher by 29%, 28%, and 26%, respectively, when compared to the 11-year average of the same months [20]. It shows that even dynamical precursors and wave activity are different at the end the effect on ozone behavior is very similar.

The difference between wave activities is possibly the main reason why SSW 2002 penetrates much deeper downward into the atmosphere than SSW 2019. This problem needs to be studied in more detail, which is out of the scope of this paper. On the other hand, as we pointed out above, the definition for major SSW from WMO is not the only one and there should be wide discussion, which definition should be used. Many studies are focused on SSW definition, e.g., [36–38]. Jucker and Reichler [39] discussed as a possible precursor Eliassen-Palm flux (EP) flux, or [40] studied problem of SSW definition and its interpretation in different model outputs. Ghosh et al. [41] shows another possibility how to study SSW. They use the vertical wavenumber (VWN) characteristics during SSW events using temperature observation of sounding of the atmosphere using broadband emission radiometry (SABER) on-board thermosphere ionosphere mesosphere energetics dynamics (TIMED) satellite. Our results show that basic characteristics like temperature or zonal wind are in principle sufficient for stratosphere but we have to look for new precursors for impact of stratospheric processes on others parts of the atmosphere.

## 5. Conclusions

SSW on the SH is very rare and that is why we compare main characteristics of the two biggest recorded ones (2002 as a major and 2019 as a minor) with big one on the NH (2009 as a major). The plausible cause of 2002 event is very strong activity of stationary planetary wave with zonal wave-number (ZW) 2, which reached maximum when the polar vortex split into two circulation cells with polar temperature enhancement by 30 K/week, and it penetrated deeply to the lower stratosphere and upper troposphere. The wave number 2 increase was initiated by south-eastward Rossby wave-trains propagated from enhanced convection regions nearby south-eastern Africa and southern Indonesia, as it was shown by [42–44]. The SSW in 2019 is only of minor type. Although it was a minor warming, it involved an exceptionally strong zonal wave-1 planetary wave and a large polar temperature enhancement by 50.8 K/week, and it affected mainly middle and upper stratosphere. We showed the difference between behavior of major SSW on the NH and SH.

The comparison of two events on the SH shows that even though the 2019 SSW was almost twice time stronger in terms of the maximal temperature enhancement, it has to be classified as minor according to the WMO definition. This brings question if this definition should not be changed or replaced. On the other hand, the type of the SSW is not so important, because we can compare effects on the middle atmosphere anyway. This study shows the basic comparison and more detail study especially for the upper levels is needed. Obtained results are based on analysis of three remarkable SSW events (2009, 2002, and 2019). Main results are as follows:

(1)  SSW in the NH affects lower stratosphere and upper troposphere (200 hPa), while in the SH it reaches down to 80 hPa at maximum.

(2)   Duration of SSW in the NH is longer than in the SH and effect of enhanced temperature can be seen several weeks after maximum in NH while in SH the normal situation recovers approximately two weeks after maximum.

(3)   The comparison of obtained results (SSW in 2002 and 2019) confirms the necessity of extension of the present WMO major SSW event definition as several studies suggested.

**Supplementary Materials:** The following are available online at http://www.mdpi.com/2073-4433/11/10/1063/s1, Figure S1: Polar temperature at 90° S for 1 and 10 hPa (upper panels) and zonal mean of zonal wind at 60° S (bottom panels) for 1 and 10 hPa during 1 August 2015–30 October 2015, Figure S2: Polar temperature at 90° N for 1 and 10 hPa (upper panels) and zonal mean of zonal wind at 60° N (bottom panels) for 1 and 10 hPa during 1 January 2014–31 March 2014, Figure S3: Polar temperature at 90° N for 1 and 10 hPa (upper panels) and zonal mean of zonal wind at 60° N (bottom panels) for 1 and 10 hPa during 1 January 2020–31 March 2020, Figure S4: Zonal mean of temperature (upper panels) and zonal wind (bottom panels) for 7, 10, and 30 hPa during period 1 August 2015–30 October 2015, Figure S5: Zonal mean of temperature (upper panels) and zonal wind (bottom panels) for 7, 10, and 30 hPa during period 1 January 2014–31 March 2014, Figure S6: Zonal mean of temperature (upper panels) and zonal wind (bottom panels) for 7, 10, and 30 hPa during period 1 January 2020–31 March 2020.

**Author Contributions:** Conceptualization, M.K. and J.L.; methodology, M.K.; validation, P.K.; formal analysis, M.K.; investigation, M.K.; resources, P.K.; data curation, M.K.; writing—original draft preparation, M.K.; writing—review and editing, M.K. and J.L.; visualization, M.K.; supervision, J.L.; project administration, J.L.; funding acquisition, J.L. All authors have read and agreed to the published version of the manuscript.

**Funding:** Support by the Czech Science Foundation through grant 18-01625S is acknowledged. This work was supported by ESA through Contract 4000126709/19/NL/IS "VERA". We thank to ECMWF team which produced ERA5 reanalysis.

**Conflicts of Interest:** The authors declare no conflict of interest.

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
