# Peer review of "Comparison of Key Characteristics of Remarkable SSW Events in the Southern and Northern Hemisphere"

_atmosphere, doi:10.3390/atmos11101063_

Round 1
Reviewer 1 Report
1. Title
As only three SSW events are analyzed the more relevant title is desirable, for instance: Comparison of Key Characteristics of remarkable SSW events in the Southern and Northern Hemisphere
2. p.1-2
The other interesting point of major SSW event definition (that could be mentioned in Introduction)
is a duration of easterly zonal-mean zonal winds: the zonal mean zonal wind reversal could be short (about 1 day) and rather weak (with values only slightly below zero).
It confirms the conventionality of difference between SSW major and SSW minor events as the last could be very strong (Manney G., et al. A minor sudden stratospheric warming with a major impact: Transport and polar processing in the 2014/2015 Arctic winter. — Geophys. Res. Lett., 2015, vol. 42, 7808—7816.)
3. p.11 lines 271-272
In relation to major SSW event in Antarctica in Sep 2002 after the following sentence "The plausible cause of 2002 event is very strong activity of stationary planetary wave with zonal wave-2 number (ZW) 2, ...."
I suggest to add somewhere that the wave number 2 increase was initiated by south-eastward Rossby wave-trains propagated from enhanced convection regions nearby south-eastern Africa and southern Indonesia as it was shown by:
Nishii K. and Nakamura H. Tropospheric influence on the diminished Antarctic ozone hole in September 2002. Geophys.Res. Lett. 2004, 31, L16103.
Peters D., Vargin P. and Koernich H. A study of the zonally asymmetric tropospheric forcing of the austral vortex splitting during September 2002. Tellus A. 2007, 59, 384[1]394.
Peters D., and Vargin P. Influence of subtropical Rossby wave trains on planetary wave activity over Antarctica in September 2002. Теllus, 2015, V. 67, 25875.
4. p.12 lines 284-289
Take into account the difference between SSW events I suggest to add to the last sentence of Conclusion that obtained results based on analysis of three remarkable SSW events
5. Following conclusion "The comparison of SSW in 2002 and 2019 brings question about SSW definition" is rather evident as this point was discussed recently in a number of papers. Moreover some ideas of new major SSW event definitions were suggested, for instance:
- Savenkova, E.N., Gavrilov, N.M., Pogoreltsev, A.I. On statistical irregularity of stratospheric warming occurrence during northern winters. J. of Atmospheric and Solar-Terrestrial Physics, 2017, V. 163, pp. 14-22
- Kodera K., Mukougawa H., Maury P., Ueda M., Claud C. Absorbing and reflecting sudden stratospheric warming events and their relationship with tropospheric circulation. J. Geophys. Res. Atmos., 2016, vol. 121, pp. 80–94
- Runde T., Dameris M., Garny H., Kinnison D. Classification of stratospheric extreme events according to their downward propagation to the troposphere. — Geophys. Res. Lett., 2016, vol. 43, pp. 6665–6672.
Therefore I suggest to modify this conclusion and write something like "..comparison of obtained results confirms the necessity of extension of the present WMO major SSW event definition"
6. I would suggest also to note in the Introduction two following recent papers on Antarctica SSW 2019:
- Milinevsky G., Evtushevsky O., Klekociuk A., Wang Y., Grytsai A., Shulga V. & Ivaniha O. Early indications of anomalous behaviour in the 2019 spring ozone hole over Antarctica, International Journal of Remote Sensing, 2020, 41:19, 7530-7540, DOI: 10.1080/2150704X.2020.1763497
- Safieddine, S., Bouillon, M., Paracho, A.‐C., Jumelet, J., Tencé, F., Pazmino, A., et al. (2020). Antarctic ozone enhancement during the 2019 sudden stratospheric warming event. Geophysical Research Letters, 47, e2020GL087810.
7. Reference 26. Gloria L. Manney, G. L., M. => Manney, G. L.,
8. Finally, the list of references is written firstly in alphabet order but later in the other way. However according to https://www.mdpi.com/journal/atmosphere/instructions#preparation - References must be numbered in order of appearance in the text.
Author Response
- Title
As only three SSW events are analyzed the more relevant title is desirable, for instance: Comparison of Key Characteristics of remarkable SSW events in the Southern and Northern Hemisphere
It has been changed as recommended.
- p.1-2
The other interesting point of major SSW event definition (that could be mentioned in Introduction)
is a duration of easterly zonal-mean zonal winds: the zonal mean zonal wind reversal could be short (about 1 day) and rather weak (with values only slightly below zero).
It confirms the conventionality of difference between SSW major and SSW minor events as the last could be very strong (Manney G., et al. A minor sudden stratospheric warming with a major impact: Transport and polar processing in the 2014/2015 Arctic winter. — Geophys. Res. Lett., 2015, vol. 42, 7808—7816.)
We have implemented this point into introduction part.
- p.11 lines 271-272
In relation to major SSW event in Antarctica in Sep 2002 after the following sentence "The plausible cause of 2002 event is very strong activity of stationary planetary wave with zonal wave-2 number (ZW) 2, ...."
I suggest to add somewhere that the wave number 2 increase was initiated by south-eastward Rossby wave-trains propagated from enhanced convection regions nearby south-eastern Africa and southern Indonesia as it was shown by:
Nishii K. and Nakamura H. Tropospheric influence on the diminished Antarctic ozone hole in September 2002. Geophys.Res. Lett. 2004, 31, L16103.
Peters D., Vargin P. and Koernich H. A study of the zonally asymmetric tropospheric forcing of the austral vortex splitting during September 2002. Tellus A. 2007, 59, 384[1]394.
Peters D., and Vargin P. Influence of subtropical Rossby wave trains on planetary wave activity over Antarctica in September 2002. Теllus, 2015, V. 67, 25875.
We have added this information including references into discussion part.
- p.12 lines 284-289
Take into account the difference between SSW events I suggest to add to the last sentence of Conclusion that obtained results based on analysis of three remarkable SSW events
We have added this information into conclusion section.
- Following conclusion "The comparison of SSW in 2002 and 2019 brings question about SSW definition" is rather evident as this point was discussed recently in a number of papers. Moreover some ideas of new major SSW event definitions were suggested, for instance:
- Savenkova, E.N., Gavrilov, N.M., Pogoreltsev, A.I. On statistical irregularity of stratospheric warming occurrence during northern winters. J. of Atmospheric and Solar-Terrestrial Physics, 2017, V. 163, pp. 14-22
- Kodera K., Mukougawa H., Maury P., Ueda M., Claud C. Absorbing and reflecting sudden stratospheric warming events and their relationship with tropospheric circulation. J. Geophys. Res. Atmos., 2016, vol. 121, pp. 80–94
- Runde T., Dameris M., Garny H., Kinnison D. Classification of stratospheric extreme events according to their downward propagation to the troposphere. — Geophys. Res. Lett., 2016, vol. 43, pp. 6665–6672.
Therefore I suggest to modify this conclusion and write something like "..comparison of obtained results confirms the necessity of extension of the present WMO major SSW event definition"
We have changed this part according to your suggestion.
- I would suggest also to note in the Introduction two following recent papers on Antarctica SSW 2019:
- Milinevsky G., Evtushevsky O., Klekociuk A., Wang Y., Grytsai A., Shulga V. & Ivaniha O. Early indications of anomalous behaviour in the 2019 spring ozone hole over Antarctica, International Journal of Remote Sensing, 2020, 41:19, 7530-7540, DOI: 10.1080/2150704X.2020.1763497
- Safieddine, S., Bouillon, M., Paracho, A.‐C., Jumelet, J., Tencé, F., Pazmino, A., et al. (2020). Antarctic ozone enhancement during the 2019 sudden stratospheric warming event. Geophysical Research Letters, 47, e2020GL087810.
We have added these two references.
- Reference 26. Gloria L. Manney, G. L., M. => Manney, G. L.,
Correted.
- Finally, the list of references is written firstly in alphabet order but later in the other way. However according to https://www.mdpi.com/journal/atmosphere/instructions#preparation - References must be numbered in order of appearance in the text.
References are now ordered as requested.
Reviewer 2 Report
This work compares SSWs between northern and southern hemispheres (NS andSH). The main flaw is that the sampling or case selection makes the results
are biased. In the past two decades, more than ten SSWs occurred in NH but
only two in the SH. The authors select the gravest NH ones to compare
with the only two from the SH, yielding biased results. Namely, the
difference between the NH and SH SSWs reported in the current work reflects
the difference between strong SSWs and weak ones rather than the difference
between NH and SH. Acutely, the most results, if not all, on the NH SSWs
have been known and therefore read redundancy.
Regarding the results on the SH SSWs, the current version presents the
most basic parameters from the model. I would encourage the authors to
do a little spectral analysis, to quantify the W1 and W2, and also to
estimate their frequencies. With these spectral analyses, I believe the
paper would be much more interesting. For the revision, I would also
encourage the authors to compare only the SH 2002 case with the SH 2019
cases. Constrain the comparison on the SH SSWs would make the main
results are more visible.
Author Response
This work compares SSWs between northern and southern hemispheres (NS and
SH). The main flaw is that the sampling or case selection makes the results
are biased. In the past two decades, more than ten SSWs occurred in NH but
only two in the SH. The authors select the gravest NH ones to compare
with the only two from the SH, yielding biased results. Namely, the
difference between the NH and SH SSWs reported in the current work reflects
the difference between strong SSWs and weak ones rather than the difference
between NH and SH. Acutely, the most results, if not all, on the NH SSWs
have been known and therefore read redundancy.
A: We are aware of possible problem with difference occurrence of SSWs on the both hemispheres.
On the other hand SSW on the SH is defined as a major (at least in 2002). This brings us to one of the main question of our paper, the definition criteria of SSW. Moreover for example temperature increase during SSW in 2019 is comparable to SSW in 2009. The other two reviewers recommend compare SSWs from SH and NH to show the difference.
Regarding the results on the SH SSWs, the current version presents the most basic parameters from the model.
I would encourage the authors to do a little spectral analysis, to quantify the W1 and W2, and also to
estimate their frequencies. With these spectral analyses, I believe the paper would be much more interesting.
A: Thank you for your recommendation but we think that this analysis is out of the scope of this paper. We are going to do more detail analysis (including spectral analysis) in the future but including more SSWs into such a study to have some statistical significance.
For the revision, I would also encourage the authors to compare only the SH 2002 case with the SH 2019 cases. Constrain the comparison on the SH SSWs would make the main results are more visible.
A: Other two reviewers recommend compare SSWs from SH and NH to show the difference between hemispheres.
Reviewer 3 Report
This is an interesting manuscript that compares two rare sudden stratospheric warming events in the Southern Hemisphere with each other and with an extreme event in the Northern Hemisphere. It can be accepted for publication following mostly minor revisions, listed below.
- Insert "zonal" before "mean" in lines 33 and 34 to distinguish it from temporal averaging.
- Change "or" to "and" in lines 43 and 44, and "studied in various papers" to "studied, for example,".
- Cite a reference for "unpredictably" in line 49, or else delete that word.
- Does reference 17 in line 56 refer to the Southern Hemisphere, as do the other references in this paragraph?
- Replace "normal" in line 70 with something like "antecedent and recovery" because these are not necessarily climatological conditions.
- Break up the long paragraph beginning on line 97 into several shorter paragraphs, each one describing a different figure. These new paragraphs would begin on lines 104, 113 and 121.
- Change "significant" to something like "substantial" in line 118 because you are not assessing statistical significance.
- Similar to comment #6, start new paragraphs on lines 146, 152 and 184.
- Change "normal" to "undisturbed" in line 203 because it is not necessarily climatological conditions.
- It is difficult to compare figures 7, 9 and 11 because their temperature scales are different. Either make the scales the same or, preferably, show temperature anomalies (departures from normal) with the same color scales.
- Change "the later" to "other" in line 228 because these studies were in the same year (2004).
- Change "brings" to "brought" in line 240 because you are discussing one case, not making a general statement.
Author Response
This is an interesting manuscript that compares two rare sudden stratospheric warming events in the Southern Hemisphere with each other and with an extreme event in the Northern Hemisphere. It can be accepted for publication following mostly minor revisions, listed below.
- Insert "zonal" before "mean" in lines 33 and 34 to distinguish it from temporal averaging.
It has been changed.
- Change "or" to "and" in lines 43 and 44, and "studied in various papers" to "studied, for example,".
It has been changed.
- Cite a reference for "unpredictably" in line 49, or else delete that word.
It has been changed.
- Does reference 17 in line 56 refer to the Southern Hemisphere, as do the other references in this paragraph?
It has been moved into another paragraph.
- Replace "normal" in line 70 with something like "antecedent and recovery" because these are not necessarily climatological conditions.
It has been changed.
- Break up the long paragraph beginning on line 97 into several shorter paragraphs, each one describing a different figure. These new paragraphs would begin on lines 104, 113 and 121.
It has been changed.
- Change "significant" to something like "substantial" in line 118 because you are not assessing statistical significance.
It has been changed.
- Similar to comment #6, start new paragraphs on lines 146, 152 and 184.
It has been changed.
- Change "normal" to "undisturbed" in line 203 because it is not necessarily climatological conditions.
It has been changed.
- It is difficult to compare figures 7, 9 and 11 because their temperature scales are different. Either make the scales the same or, preferably, show temperature anomalies (departures from normal) with the same color scales.
The figures have the same scales. It was the wrong scale bar on Fig.7 but now it is corrected. We prefer real temperature because it shows real conditions without any artificial input.
- Change "the later" to "other" in line 228 because these studies were in the same year (2004).
It has been changed.
- Change "brings" to "brought" in line 240 because you are discussing one case, not making a general statement.
It has been changed.
Round 2
Reviewer 2 Report
I do not think the manuscript has visible improvement in comprasion with the last version.
Author Response
This work compares SSWs between northern and southern hemispheres (NS and SH). The main flaw is that the sampling or case selection makes the results are biased. In the past two decades, more than ten SSWs occurred in NH but only two in the SH. The authors select the gravest NH ones to compare with the only two from the SH, yielding biased results. Namely, the difference between the NH and SH SSWs reported in the current work reflects the difference between strong SSWs and weak ones rather than the difference between NH and SH. Acutely, the most results, if not all, on the NH SSWs have been known and therefore read redundancy.A: We are aware of possible problem with difference occurrence of SSWs on the both hemispheres and we added several sentences about this. On the other hand SSW on the SH is defined as a major (at least in 2002).
This brings us to one of the main question of our paper, the
definition criteria of SSW. Moreover for example temperature increase
during SSW in 2019 is comparable to SSW in 2009. The other two
reviewers strongly recommend compare SSWs from SH and NH to show the difference and it follows from their reviews that comparison using only SH SSWs can reduce the level of this paper.
Regarding the results on the SH SSWs, the current version presents the most basic parameters from the model.
I would encourage the authors to do a little spectral analysis, to
quantify the W1 and W2, and also to estimate their frequencies. With these spectral analyses, I believe
the paper would be much more interesting.
A: Thank you for your recommendation but we think that this analysis
is out of the scope of this paper. Using spetral analysis for two SSWs
without possible comparison to more events can bring biased results.
We are going to do more detail analysis (including spectral analysis)
in the future but including more SSWs into such a study to have some
statistical significance. But we have added several paragraphs
including discussion of wave analysis and ozone behaviour during SSW
2002 and 2019 with several additional references to shed some light on the problem.
For the revision, I would also encourage the authors to compare only
the SH 2002 case with the SH 2019 cases. Constrain the comparison on the SH SSWs would make the main results are more visible.
A: Other two reviewers strongly recommend compare SSWs from SH and NH to show the difference between hemispheres, and it follows from their reviews that comparison using only SH SSWs can reduce the level of this paper.